# T2D: Spatiotemporal Feature Learning Based on Triple 2D Decomposition

## Abstract

In this paper, we propose triple 2D decomposition (T2D) of a 3D vision Transformer (ViT) for efficient spatiotemporal feature learning. The idea is to decompose the self-attention operation in a 3D data cube into three self-attention operations in three 2D data planes. Such a design not only effectively reduces the computational complexity of a 3D ViT, but also guides the network to focus on learning correlations among more relevant tokens. Compared with other decomposition methods, the proposed T2D is shown to be more powerful at a similar computational complexity. The CLIP-initialized T2D-B model achieves state-of-the-art top-1 accuracy of 85.0% and 70.5% on Kinetics-400 and Something-Something-v2 datasets, respectively. It also outperforms other methods by a large margin on FineGym (+17.9%) and Diving-48 (+1.3%) datasets. Under the zero-shot setting, the T2D model obtains a 2.5% top-1 accuracy gain over X-CLIP on HMDB-51 dataset. In addition, T2D is a general decomposition method that can be plugged into any ViT structure of any model size. We demonstrate this by building a tiny size of T2D model based on a hierarchical ViT structure named DaViT. The resulting DaViT-T2D-T model achieves 82.0% and 71.3% top-1 accuracy with only 91 GFLOPs on Kinectics-400 and Something-Something-v2 datasets, respectively. Source code will be made publicly available.

## 1 Introduction

Learning spatiotemporal representation for videos is one of the most fundamental yet challenging tasks in computer vision (CV). The challenges come mainly from the contradiction between the insufficient data and the diverse spatiotemporal patterns that need to be learned. It will become very obvious if we take the representation learning for images as a reference. The largest public image dataset ImageNet-21K Deng et al. (2009) consists of more than 14M images divided into over 21K classes, while the largest public video dataset Kinetics-700 Carreira et al. (2019) only consists of around 500K videos divided into 700 human action classes. Video data are far inferior to image data in terms of quantity and diversity, but the spatiotemporal information to be learned is one dimension higher than the spatial information contained in the image.

A straightforward idea to address this challenge in video representation learning is to make full use of the spatial modeling capability gained by the image models. Researchers started to implement this idea back in the convolution neural network (CNN) era. For example, I3D network Carreira & Zisserman (2017) uses inflated ResNet weights trained on ImageNet for initialization. More recently, as the Transformer Vaswani et al. (2017) architecture starts to dominate in CV, the ViT Dosovitskiy et al. (2021) network pretrained on ImageNet or by CLIP Radford et al. (2021) has been adopted as the initialization of spatiotemporal feature learning networks in many efforts Wang et al. (2021b); Pan et al. (2022); Lin et al. (2022).

The quadratic complexity of Transformer brings a great challenge to the adaptation from image-oriented 2D ViT to video-oriented 3D ViT. Decomposition is in need to make the computation complexity tractable. Previous work Bertasius et al. (2021) has proposed space-time decomposition, axial decomposition, or local-global decomposition, among others, but they have not achieved satisfactory performance. In fact, the key question to be considered when decomposing a 3D ViT is which tokens should be grouped together to perform the self-attention. The selection of group size and coverage should trade off the computational complexity and the attainable modeling capability.

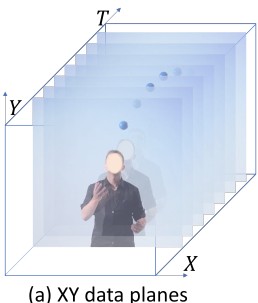 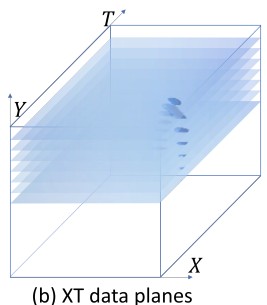 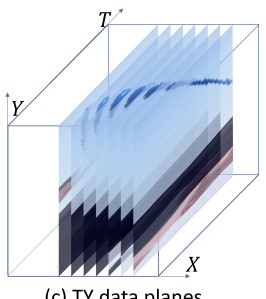

(a) XY data planes        (b) XT data planes        (c) TY data planes

Figure 1: Decomposing 3D video data XYT into three data planes, denoted by XY, XT, and TY. The XY data plane is sufficient for recognizing main objects. The XT and TY data planes provide rich information of object motion.

We propose triple 2D decomposition for effective spatiotemporal learning from videos, which is a third-order tensor. Decomposition of higher-order tensors has found many applications in computer vision to deal with 3D geometry Chan et al. (2022); Chen et al. (2022a) or video modeling Tran et al. (2018); Bertasius et al. (2021). Our work shares a similar idea as the tri-plane representation in EG3D Chan et al. (2022) to decompose a 3D tensor into three 2D data planes. Let X, Y, and T denote the horizontal, vertical, and temporal axis of a video tensor, respectively. The XY data plane contains sufficient spatial information for recognizing the main objects, and the two extended temporal data planes XT and TY provide rich information of object motions, as Fig.1 shows. The T2D design groups the tokens in the same XY, XT, or TY plane for self-attention computation. The group size is more manageable than the default 3D attention. More importantly, all the computational resources and the available training data are spent on mining the correlation of the most relevant tokens. Besides, we propose to share weights between the XT and TY data planes, while leaving the XY branch to be separately initialized with pretrained weights of an image model.

The proposed triple 2D decomposition is a straightforward design, but not sufficiently explored by previous researchers. We think the possible reasons are twofold. First, previous research mainly focused on reducing computational complexity of models, and such T2D decomposition do not reduce the complexity in a typical CNN setting[1]. Second, the action recognition performance is usually evaluated on simple or scene-focused datasets, such as UCF-101 Kuehne et al. (2011), HMDB-51 Soomro et al. (2012), and Kinetics Kay et al. (2017), where spatial modeling plays a dominant role. In this case, temporal modeling has not received the attention it deserves. In summary, the main contributions of our work are three-fold:

- We propose triple 2D decomposition for efficient spatiotemporal feature learning with a Transformer network. Isolating self-attention operation within each 2D data plane not only makes the computational complexity easily manageable, but offers great design flexibility, allowing us to select different settings and initialization for spatial and temporal modeling.

- We provide a detailed analysis of different decomposition methods for 3D ViT and carry out ablation studies to demonstrate the advantage of the proposed T2D.

- We make T2D a plug-n-play component and implement it based on both CLIP ViT Radford et al. (2021) and DaViT Ding et al. (2022). All versions of the T2D network achieve higher or competitive performance on Kinetics-400 and Something-Something-v2 benchmarks when compared with state-of-the-art (SOTA) models of similar sizes. The CLIP-based T2D network is extensively evaluated on a broad range of video action recognition benchmarks. T2D-B pushes previous SOTA from 88.0%/86.4%/50.9% to 89.3%/93.6%/68.8% on Diving-48, Gym99, and Gym288, respectively. It also achieves competitive or higher performance in zero-shot evaluation on HMDB-51 and UCF-101 datasets compared to previous SOTA ActionCLIP Wang et al. (2021b) and X-CLIP Ni et al. (2022).

---

[1] A typical $3 \times 3 \times 3$ convolution kernel has the same computation complexity compared with three $3 \times 3$ convolution kernels.

## 2 RELATED WORK

Videos are moving pictures. Video models learn spatiotemporal representation from the 3D input while image models only learn spatial representation from the 2D input. Whether from a semantic or operational level, a video model should leverage the more intensely studied image models. In this section, we review related work on spatiotemporal feature learning from two perspectives. One is how a video model is factorized to make use of the 2D operations, and the other is how a video model can be built upon the spatial modeling capabilities of image models.

**Factorized Video Model Design.** A video model extracts spatiotemporal features from the input 3D data. It is natural to use the 3D kernel as the building block of a video model Tran et al. (2015); Arnab et al. (2021). Even in the CNN era, 3D kernels suffer from high computational and memory cost Tran et al. (2018). Cost reduction can be approached by replacing part of the 3D kernels with 2D kernels or factorizing a 3D kernel (e.g., $3 \times 3 \times 3$) into a 2D spatial kernel (e.g., $1 \times 3 \times 3$) and a 1D temporal kernel (e.g., $3 \times 1 \times 1$). P3D Qiu et al. (2017), R(2+1)D Tran et al. (2018), and S3D Tran et al. (2018) are contemporaneous works that explore factorization CNNs.

Recently, with the widespread adoption of vision Transformers (ViT) Dosovitskiy et al. (2021); Dong et al. (2022); Chen et al. (2022b); Wang et al. (2022b); Ding et al. (2022), the complexity issue in video models becomes more prominent. As is well known, the attention operation in Transformers Vaswani et al. (2017) has quadratic complexity with respect to the input token numbers. If we use $S$ to denote the spatial resolution and $T$ the temporal resolution, the full space-time attention has the complexity of $O(T^2S^4)$. TimeSformer Bertasius et al. (2021) explores the divided spatial and temporal attention with the complexity of $O(TS^2(T + S^2))$ to replace the full space-time attention. It is actually the Transformer version of R(2+1)D. In a similar fashion, ViViT Arnab et al. (2021) also explores several space-time factorization mechanisms for video Transformer. We will provide a detailed analysis of different decomposition methods for 3D ViT in Section 3.2.

**Video Models Built Upon Pretrained ViT Models.** It has been a common practice since the CNN era to initialize a video model with the weights from a pretrained image model Carreira & Zisserman (2017); Arnab et al. (2021); Liu et al. (2021b). While the 3D kernels in CNN need to be inflated from 2D kernels, the token-based Transformer architecture can be more easily converted from a image model to a video model. For example, ViViT and TimeSformer are initialized with the weights of ViT, which are pretrained by the supervised image classification task on ImageNet-21K. For another example, Video Swin Transformer Liu et al. (2021b) uses the weights from Swin Transformer, also pretrained on ImageNet-21K.

Beyond the ImageNet pretrained model, a surge of new image foundation models Radford et al. (2021); Yuan et al. (2021); Yu et al. (2022) for general visual representation learning have been developed with the availability of large-scale weakly labeled image-text data. Such image-text pretrained models has demonstrated impressive generalization capacities and even "zero-shot" transfer capability. CLIP Radford et al. (2021), as one representative work, has already been extended to video models in many efforts. Text4Vis Wu et al. (2022) proposed a new visual tuning paradigm by leveraging the textual knowledge from CLIP. Concurrently, X-CLIP Ni et al. (2022) proposed video-specific prompting to better utilize the text information. It is found in other works Lin et al. (2022); Wang et al. (2021b) that frozen CLIP models could already achieve satisfactory performance so efficient transfer learning pipelines are built to reduce the transfer cost.

We build our main T2D network based on CLIP. The CLIP-version of our T2D network can leverage the accompanying text encoder just as X-CLIP does and it has the zero-shot adaptation capability. Yet, we point out that T2D is a general decomposition method which can be implemented in any ViT-based network structures. To demonstrate its plug-n-play capability, we also implement a version based on another ImageNet-21K pretrained image Transformer model DaViT Ding et al. (2022).

## 3 METHOD

We build a video model, named T2D network, for efficient spatiotemporal feature learning and video action recognition. In this section, we first provide an overview of the proposed T2D network, and then analyze the core component, namely T2D decomposition, through the comparison with previous decomposition methods.

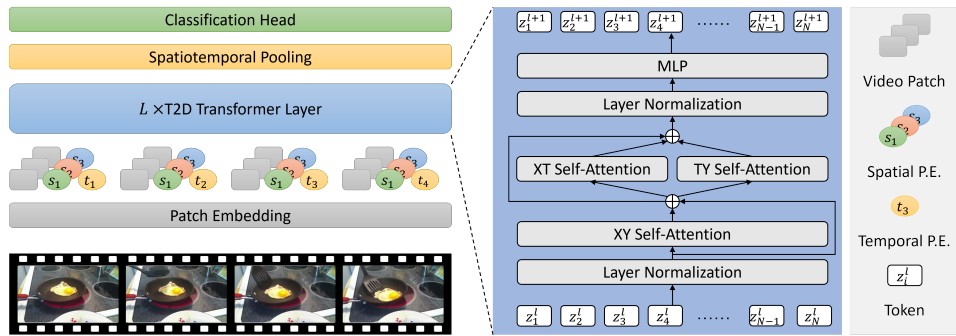

Figure 2: Overview of the proposed T2D network for video action recognition and the implementation details of the T2D Transformer layer. P.E. is short for positional embedding.

## 3.1 OVERVIEW

The T2D network is a video vision Transformer adapted from ViT Dosovitskiy et al. (2021) to process 3D videos. Its architecture is illustrated in Fig.2. Let us denote the input video clip by $X \in \mathbb{R}^{T \times H \times W \times 3}$, where $H$ and $W$ are the height and width of video frames, $T$ is the clip length, and 3 is the number of RGB channels. The T2D network first divides the input video clip into $N$ non-overlapping patches, each of size $P_t \times P_s \times P_s$, where $P_t$ and $P_s$ are the temporal and the spatial patch size, respectively, and embeds them into $X_p \in \mathbb{R}^{N \times C}$, where $C$ is the number of channels. Then positional embedding is added to obtain the input $Z^0$ to the first T2D Transformer layer:

$$Z^0 = X_p + \boldsymbol{e}^s + \boldsymbol{e}^t, \tag{1}$$

where $e^s$ and $e^t$ are learnable spatial and temporal positional embeddings, respectively. The '+' here represents broadcast addition operation, which is explained in detail in Appendix.

The core network is composed of $L$ T2D Transformer layers, whose structure is also shown in Fig.2. Each T2D Transformer layer is composed of Multi-Headed Self-Attention (MSA) blocks, layer normalization (LN) layers, and the multi-layer perceptron (MLP) blocks. Let $Z^{l-1}$ and $Z^l$ denote the input and the output of the $l^{th}$ Transformer layer, respectively, and the computation implemented by this layer can be written as:

$$Y^l = \text{MSA}(\text{LN}(Z^{l-1})) + Z^{l-1} \tag{2}$$
$$Z^l = \text{MLP}(\text{LN}(Y^l)) + Y^l. \tag{3}$$

Finally, spatiotemporal pooling is performed on the output $Z^L$ of the last Transformer layer, and a linear classifier is attached to classify the video features into predefined categories.

## 3.2 T2D DECOMPOSITION FOR 3D VIT

Decomposition for 3D ViT is necessary for two reasons. First, directly computing self-attention among all tokens in a 3D cube incurs huge computational cost Vaswani et al. (2017). Second, the quantity and diversity of existing video training data are limited. Considering the complexity and diversity of the spatiotemporal information to be learned, it is possible that we never get enough data to fully explore the capacity of a 3D ViT. In this case, we shall prioritize training on the most relevant set of tokens. This serves as the motivation of our T2D proposal.

Fig.3 illustrates how a patch in frame $T$ interacts with adjacent patches in different decomposition methods. In addition to the proposed T2D decomposition, we also include the space-only (2D) attention, the divided space-time (2D+1D) attention, the local-global attention, and the axial (1D+1D+1D) attention mentioned in previous work Bertasius et al. (2021). The implementation of these decomposition methods are described below:

**Space-only (2D) attention.** Attention is performed within each frame but not across frames. The advantage is that the computational complexity is exactly the same as in 2D image ViT, but the

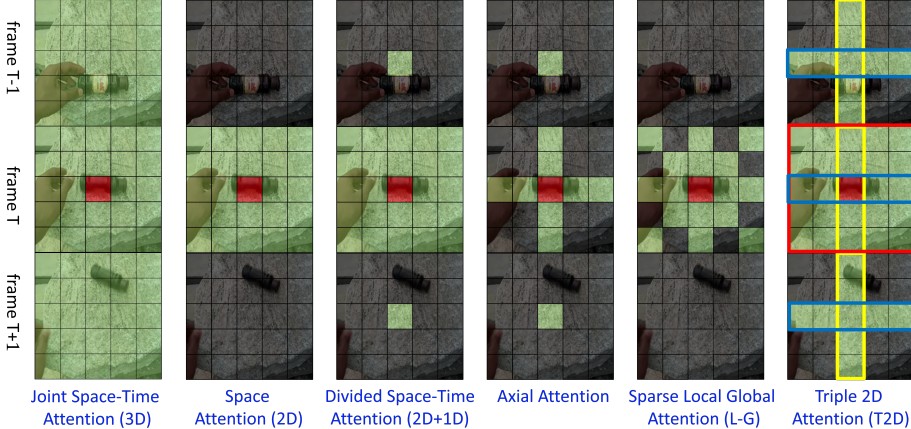

Figure 3: Visualization of different decomposition schemes. We denote the query patch as filled red and its attention neighborhood in green. Uncorrelated patches are masked in black. We show the center frame and its two adjacent frames. The proposed T2D attention, where each colored border represents a single attention plane (XY: red, XT: blue, TY: yellow), explores the correlation in a wider range than other methods under comparable complexity.

shortcoming is also obvious. The temporal modeling based on simple pooling operation across multiple frames is too rudimentary to achieve good performance on motion-focused datasets.

**Divided space-time (2D+1D) attention.** A more widely adopted structure is 2D+1D decomposition. This approach first computes the self-attention within each frame, and then compute the self-attention along the temporal axis among tokens from the same spatial index. This approach also has controllable complexity, but temporal attention without spatial resolution might fail to catch objects in other frames, as the example shows in Fig.3.

**Axial (1D+1D+1D) attention.** The axial attention further decomposes the 2D spatial attention into two 1D attention operations. The computational complexity is further reduced when compared with divided space-time attention, but the spatial modeling capability is also compromised.

**Local-global attention.** Another less frequently mentioned approach is local-global decomposition Child et al. (2019). The local self-attention is first computed in the $N_t \times N_h/2 \times N_w/2$ local windows, and the global sparse self-attention is then calculated over the entire clip with a stride of 2 tokens along both temporal and spatial dimensions. Compared to divided space-time attention, the local-global attention sacrifices some spatial modeling capability for stronger temporal modeling.

**T2D attention** Finally, the last column in Fig.3 illustrates the proposed T2D decomposition. Self-attention is computed among patches within red, blue, and yellow boxes, which represent the XY, XT, and TY data planes, respectively. The computation complexity is only slightly higher than divided space-time attention, but T2D decomposition explores the patch correlations in a much wider range than other decomposition methods. T2D attention can be easily implemented by feature reshaping. Take the attention in TX data plane as an example. An input $Z \in \mathbb{R}^{1 \times (N_h \cdot N_w \cdot N_t) \times C}$ can be reshaped to $Z_{tx} \in \mathbb{R}^{N_h \times (N_w \cdot N_t) \times C}$ before computing the pair-wise self-attention.

Table 1 outlines the per-layer complexity of the MSA module in different decomposition methods.

Table 1: Per-layer complexity of the MSA module in different decomposition schemes.

| Method | Complexity |
|---|---|
| 2D Attention | $\mathcal{O}(N_t N_h^2 N_w^2)$ |
| 2D+1D Attention | $\mathcal{O}(N_t N_h^2 N_w^2 + N_t^2 N_h N_w))$ |
| Axial Attention | $\mathcal{O}(N_t N_h N_w^2 + N_t N_h^2 N_w + N_t^2 N_h N_w)$ |
| L-G Attention | $\mathcal{O}(\frac{5}{64} N_t^2 N_h^2 N_w^2)$ |
| T2D Attention | $\mathcal{O}(N_t N_h^2 N_w^2 + N_t^2 N_h^2 N_w + N_t^2 N_h N_w^2)$ |

# 4 EXPERIMENTS

## 4.1 EXPERIMENTAL SETUP

**Architecture.** In main experiments, we use ViT Dosovitskiy et al. (2021) as the base model. In T2D Transformer layers, the added XT and TY self-attention use shared parameters and they are sequentially connected after the original XY self-attention. We use CLIP pre-trained weights to initialize the MLP and XY self-attention blocks. Other parameters are randomly initialized. We build two variants T2D-B and T2D-L, derived from ViT-B/16 and ViT-L/14, respectively. To demonstrate that T2D is a plug-n-play block, we also build a tiny-size model called DaViT-T2D-T from a hierarchical Transformer DaViT-T Ding et al. (2022) as well as a CNN-based model ResNet-T2D from ResNet.

**Benchmark datasets and evaluation protocols.** We evaluate the proposed T2D network under fine-tuning protocol on major action recognition benchmark datasets, including Kinetics-400 (K400) Kay et al. (2017), Something-Something-v2 (SSv2) Goyal et al. (2017), FineGym Shao et al. (2020), and Diving-48 Li et al. (2018). We also report zero-shot results of the CLIP-initialized T2D network on UCF-101 Soomro et al. (2012) and HMDB-51 Kuehne et al. (2011) datasets.

**Training and inference.** For all datasets, we use $224 \times 224$ input clips with clip length of 16 or 32. We follow the sparse sampling and data augmentation in X-CLIP Ni et al. (2022) on K400. On SSv2, Diving-48, and FineGym datasets, we use segment-based sampling as in TSM Lin et al. (2019) and follow the augmentation in MViT Fan et al. (2021). More details are provided in Appendix.

## 4.2 COMPARISON OF DECOMPOSITION METHODS

Tab.2 presents the performance of different decomposition methods mentioned in Section 3.2 on multiple network structures and benchmark datasets. The local-global attention and axial attention are not included in this experiment as they have already been proven to be inferior to the divided space-time (2D+1D) attention Bertasius et al. (2021).

For ViT-based models, we find that different decomposition methods have similar performance on K400. Even the space-only (2D) attention achieves a high top-1 accuracy of 84.0%. This is consistent with previous findings Bertasius et al. (2021) that spatial modeling dominates the performance on K400, which is a scene-focused dataset. Nevertheless, our T2D network achieves the highest top-1 accuracy among all the methods. We also tried to remove XT or TY attention in the T2D model, and we obtain 84.3% and 84.1% top-1 accuracy, respectively, showing that all three data planes contribute to the action recognition performance. On SSv2, however, the performance of different methods varies widely. The 2D attention performs poorly on this motion-focused dataset as it lacks temporal modeling. The 3D attention performs even worse and we suspect that it is due to the lack of training data. The space-time decomposition models, including 2D+1D and T2D perform significantly better. Our T2D model achieves the best top-1 accuracy on SSv2.

We also carry out experiments to evaluate different decomposition methods on CNN-based ResNet and another Transformer-based model DaViT. We obtain very similar results with the ViT model. In order to demonstrate that the gain achieved by T2D over 2D+1D is not due to the slightly increased computational cost, we use light-weight depth-wise convolution to implement the computation for 1D in 2D+1D and that for XT and TY in T2D. The resulting ResNet-based 2D + 1D model and T2D model have almost the same GFLOPs, but T2D model still outperforms 2D+1D model by a 0.4% top-1 accuracy on K400. We can see that the proposed T2D decomposition achieves consistent non-negligible gain over divided space-time (2D+1D) decomposition across different network structures.

Table 2: Comparison of different decomposition methods based on three network structures.

| Method | ViT | | | DaViT | | ResNet | |
|---|---|---|---|---|---|---|---|
| | GFLOPs | K400@1 | SSv2@1 | GFLOPs | K400@1 | GFLOPs | K400@1 |
| 2D | 282 | 84.0 | 68.3 | 74 | 80.6 | 93 | 71.3 |
| 3D | 452 | 84.3 | 67.2 | 92 | 80.3 | 186 | 70.9 |
| 2D+1D | 372 | 84.1 | 70.2 | 86 | 80.6 | 93 | 72.3 |
| T2D | 397 | 84.5 | 70.5 | 91 | 81.0 | 94 | 72.7 |

Table 3: Ablation of T2D block.

| ID. | Con. | Share | GFLOPs | SSv2@1 |
|---|---|---|---|---|
| 1 | Seq. | Tem. | 397 | 70.5 |
| 2 | Par. | Tem. | 397 | 68.7 |
| 3 | Seq. | None | 486 | 70.6 |
| 4 | Seq. | All | 308 | 69.9 |

Table 4: Zero-shot performances on HMDB-51 and UCF-101.

| Method | HMDB-51 | UCF-101 |
|---|---|---|
| ActionCLIP Wang et al. (2021b) | $40.8 \pm 5.4$ | $58.3 \pm 3.4$ |
| X-CLIP-B/16 Ni et al. (2022) | $44.6 \pm 5.2$ | $\mathbf{72.0 \pm 2.3}$ |
| T2D-B | $\mathbf{47.1 \pm 0.3}$ | $71.1 \pm 0.2$ |

## 4.3 ABLATION STUDIES

Before diving into system comparisons with other methods, we present the ablation studies on how to connect different branches and whether to share parameters among the three branches in the T2D blocks. We use T2D-B with 32 frames input for all the ablation studies.

We explore two connection variants, one is to sequentially connect the spatial and temporal kernel with skip connection, and the other is to connect the two kernels in parallel. Here the temporal kernel means the cascade of XT and TY self-attention, and the spatial kernel means the XY self-attention. As shown in Tab.3, the sequentially connected model performs significantly better as such a structure makes the adaptation from image model to video model more naturally. We use the sequential connected model as the default setting in other experiments.

The other question in T2D is whether we should share parameters among three self-attentions. We design three parameter sharing variants. The first model shares parameters between two temporal attention blocks, the second model does not share parameters, and the third model shares parameters among all three self-attentions. The parameters shared with spatial self-attention are initialized from CLIP pre-trained model, otherwise they are randomly initialized. As shown in Tab.3, the non-shared version obtains the best performance while the temporal shared version strikes the best performance-accuracy trade-off. It is as expected that the all-shared model does not achieve plausible performance, as spatial modeling and temporal modeling have their own distinct characteristics.

It is worth noting that the all shared model is similar to the CoST method proposed by Li et al. (2019), which aims to learn spatial and temporal features collaboratively for CNN. Although they use similar decomposition formulation as ours, the insight behind such decomposition is different. Their experiments on K400 verify that the sharing design is the key to their performance improvement over 3D model. However, in our experiments, we find that collaborative spatiotemporal feature learning hurts the performance. Concretely, sharing spatial and temporal parameters introduces 0.6% top-1 accuracy loss compared to the model with temporal sharing only. Such performance loss indicates the importance of decoupling spatial and temporal feature learning.

## 4.4 COMPARISON TO THE STATE-OF-THE-ART

**Kinetics-400.** In Tab.5, we report the results on Kinetics-400 with comparison to state-of-the-art methods group by model architectures and pre-train models. Compared to methods with CNN and methods with Transformer that do not using CLIP pre-trained weights, our T2D have a large performance gain. Specifically, on base size, our T2D-B with 16 frames outperform X3D-XXL Feichtenhofer (2020) by 4.3% top-1 accuracy and Uniformer-B Li et al. (2022) by 1.7% top-1 accuracy. On large size, our T2D-L with 16 frames outperform Video-Swin-V2-G (384 ↑) Liu et al. (2021a) by 0.8% and TokenLearner by 2.2% although these two models are even heavier. Compare to methods with the same CLIP pre-training, our T2D models also show competitive performance. On base size, the T2D-B with 16 frames outperforms ActionCLIP Wang et al. (2021b) by 0.9%, EVL Lin et al. (2022) by 0.5%, ST-Adapter Pan et al. (2022) by 2.0% and Text4Vis Wu et al. (2022) by 1.1%. The T2D-B with 32 frames achieves 85.0% top-1 accuracy which is the best among all compared models. On large size, T2D-L with 32 frames achieves 87.8% top-1 accuracy and 97.9% top-5 accuracy which are the SOTA. On tiny size, DaViT-T2D-T with only 91 GFLOPs outperforms Video-Swin-S Liu et al. (2021b) by 1.4% and MTV-B Yan et al. (2022) by 0.2%.

**FineGym.** Finegym is a newly proposed dataset built with gymnastic videos which contains subtly different gymnastic actions. We compare our T2D-B with 32 frames with baseline methods established by Shao et al. (2020) and SOTA methods including SELFYNetKwon et al. (2021) and RSANet Kim et al. (2021). Results are shown in Tab.6. Surprisingly, our model improves previous

Table 5: Comparison to the state-of-the-art on Kinetics-400."#Frames" denotes the total number of frames used during inference which is #frames per clip × # spatial crop × # temporal clip.

| Method | #Frames | GFLOPs | Top-1 | Top-5 |
|---|---|---|---|---|
| *Methods with CNN* | | | | |
| R(2+1)D Tran et al. (2018) | 16x1x10 | 75 | 72.0 | 90.0 |
| CoST Tran et al. (2018) | 32x- | - | 77.5 | 93.2 |
| SlowFast + NL Feichtenhofer et al. (2019) | 16x3x10 | 234 | 79.8 | 93.9 |
| X3D-XXL Feichtenhofer (2020) | 16x3x10 | 144 | 80.4 | 94.6 |
| *Methods with Transformer* | | | | |
| Video-SwinV2-G (384 ↑) Liu et al. (2021a) | 8x5x4 | - | 86.8 | - |
| TokenLearner Ryoo et al. (2021) | 64x3x4 | 4076 | 85.4 | 96.3 |
| ViViT-L FE Arnab et al. (2021) | 32x3x1 | 3980 | 83.5 | 94.3 |
| MViTv2-L (312 ↑) Li et al. (2021) | 40x3x5 | 2828 | 86.1 | 97.0 |
| TimeSformer-L Bertasius et al. (2021) | 96x3x1 | 2380 | 80.7 | 94.7 |
| CoVeR TimeSformer-L Zhang et al. (2021) | -x3x1 | - | 87.2 | - |
| Video-Swin-L (384 ↑) Liu et al. (2021b) | 32x5x10 | 2107 | 84.9 | 96.7 |
| MTV-L Yan et al. (2022) | 32x3x4 | 1504 | 84.3 | 96.3 |
| MTV-B Yan et al. (2022) | 32x3x4 | 399 | 81.8 | 95.0 |
| Uniformer-B Li et al. (2022) | 32x3x4 | 259 | 83.0 | 95.4 |
| MViTv2-B Li et al. (2021) | 32x1x5 | 225 | 82.9 | 95.7 |
| Video-Swin-S Liu et al. (2021b) | 32x3x4 | 166 | 80.6 | 94.5 |
| DaViT-T2D-T (**Ours**) | 32x3x4 | 91 | 82.0 | 95.5 |
| *Methods with CLIP-B pre-trained ViT* | | | | |
| ActionCLIP-B/16 Wang et al. (2021b) | 32x3x10 | 563 | 83.8 | 96.2 |
| EVL ViT-B/16 Lin et al. (2022) | 32x3x1 | 592 | 84.2 | - |
| X-CLIP-B/16 Ni et al. (2022) | 16x3x4 | 287 | 84.7 | 96.8 |
| ViT-B w/ ST-Adapter Pan et al. (2022) | 32x3x1 | 607 | 82.7 | 96.2 |
| Text4Vis-B/16 Wu et al. (2022) | 16x3x4 | - | 83.6 | 96.4 |
| T2D-B (**Ours**) | 16x3x4 | 395 | 84.7 | 96.7 |
| T2D-B (**Ours**) | 32x3x4 | 842 | **85.0** | **96.8** |
| *Methods with CLIP-L pre-trained ViT* | | | | |
| EVL ViT-L/14 (336px) Lin et al. (2022) | 32x3x1 | 6065 | 87.7 | - |
| X-CLIP-L/14 (336 ↑) Ni et al. (2022) | 16x3x4 | 3086 | 87.7 | 97.4 |
| ViT-L w/ ST-Adapter Pan et al. (2022) | 32x3x1 | 2749 | 87.2 | 97.6 |
| Text4Vis-L/14 (336 ↑) Wu et al. (2022) | 32x3x1 | 3829 | 87.8 | 97.6 |
| T2D-L (**Ours**) | 16x3x4 | 1807 | 87.6 | 97.7 |
| T2D-L (**Ours**) | 32x3x4 | 3821 | **87.8** | **97.9** |

Table 6: Comparison to SOTA on FineGym.

| Method | Gym288 Mean | Gym99 Mean |
|---|---|---|
| I3D Carreira & Zisserman (2017) | 27.9 | 63.2 |
| TSM Lin et al. (2019) | 34.8 | 70.6 |
| SELFYNet Kwon et al. (2021) | 49.5 | 87.7 |
| RSANet Kim et al. (2021) | 50.9 | 86.4 |
| T2D-B | **68.8** | **93.6** |

Table 7: Comparison to SOTA on Diving48.

| Method | Top-1 |
|---|---|
| SlowFast Feichtenhofer et al. (2019) | 77.6 |
| TimseSformer-L Bertasius et al. (2021) | 81.0 |
| PST-B Xiang et al. (2022) | 86.0 |
| BEVT Wang et al. (2022a) | 86.7 |
| ORViT TimsSformer Herzig et al. (2021) | 88.0 |
| T2D-B | **89.3** |

SOTA by 17.9% and 7.2% on Gym288 and Gym99, respectively, in terms of mean class accuracy. Such strong results demonstrate the powerful temporal modeling capability of our T2D models.

**Diving-48.** Tab. 7 compares our T2D-B with SOTA methods on Diving-48, which is also a fine-grained action recognition dataset. It contains 48 classes of competitive diving sequences. We compare our method with strong baselines SlowFast Feichtenhofer et al. (2019) and TimeSformer Bertasius et al. (2021), as well as recent proposed SOTA methods PST-B Xiang et al. (2022) and ORViT TimeSformer Herzig et al. (2021). We achieve 8.3% top-1 accuracy gain over TimeSformer-L even with a smaller model size. We also outperform previous best method ORViT by 1.3%.

Table 8: Comparison to the state-of-the-art on Something-Something-v2.

| Method | #Frames | GFLOPs | Top-1 | Top-5 |
|---|---|---|---|---|
| *Methods with CNN* | | | | |
| TSM Lin et al. (2019) | 16x1x1 | 66 | 63.3 | 88.5 |
| MSNet Kwon et al. (2020) | 16x1x1 | 67 | 64.7 | 89.4 |
| SELFYNet Kwon et al. (2021) | 16x1x1 | 67 | 65.7 | 89.8 |
| TDN Wang et al. (2021a) | 16x1x1 | 132 | 66.9 | 90.9 |
| *Methods with hierarchical Transformer* | | | | |
| Video-Swin-B Liu et al. (2021b) | 32x3x1 | 321 | 69.6 | 92.7 |
| UniFormer-B Li et al. (2022) | 32x3x1 | 259 | 71.2 | 92.8 |
| MViT-B-24 Fan et al. (2021) | 32x3x1 | 236 | 68.7 | 91.5 |
| MViTv2-S Li et al. (2021) | 32x3x1 | 65 | 68.2 | 91.4 |
| MViTv2-B Li et al. (2021) | 32x3x1 | 225 | 72.1 | 93.4 |
| DaViT-T2D-T **(Ours)** | 32x3x1 | 91 | 71.3 | 93.0 |
| *Methods with cylindrical Transformer* | | | | |
| TimeSformer-HR Bertasius et al. (2021) | 16x3x1 | 1703 | 62.5 | - |
| ViViT-L Arnab et al. (2021) | 16x3x4 | 903 | 65.4 | 89.8 |
| MTV-B (320p) Yan et al. (2022) | 16x3x4 | 930 | 68.5 | 90.4 |
| Mformer-L Patrick et al. (2021) | 32x3x1 | 1185 | 68.1 | 91.2 |
| EVL ViT-B/16 Lin et al. (2022) | 32x3x1 | 682 | 62.4 | - |
| ViT-B w/ ST-Adapter Pan et al. (2022) | 32x3x1 | 652 | 69.5 | 92.6 |
| T2D-B **(Ours)** | 32x3x2 | 397 | **70.5** | **92.6** |

**Something-Something-v2.** Tab.8 presents the results of T2D and SOTA methods on SSv2. We group the methods into three categories by model architecture. The first group is CNN-based methods which do not achieve competitive performance compared to newly proposed Transformer-based methods. The second group is hierarchical Transformer-based methods. The best method MViTv2-B Li et al. (2021) achieves the highest top-1 accuracy of 72.1% with 225 GFLOPs. Our DaViT-T2D-T model also achieves a competitive top-1 accuracy of 71.3% with only 91 GFLOPs. It strikes a better performance-complexity tradeoff than MViTv2 models as it obtains 2.9% top-1 accuracy gain over MViTv2-S with only 40% more FLOPs. The third group is cylindrical Transformer-based methods which use the same model architecture as ours. Although this architecture seems to be inferior to hierarchical Transformer on SSv2 dataset, our T2D-B manages to achieve 70.5% top-1 accuracy and outperforms all competitors in this category.

**Zero-shot results on UCF-101 and HMDB-51.** Finally, we present zero-shot results of T2D in Tab.4 to demonstrate the generalization capability of our methods. We use the K-400 pre-trained models and test on UCF-101 and HMDB-51. The text encoder in CLIP is fixed during training and zero-shot testing. Details of zero-shot implementation are provided in the Appendix. T2D outperforms previous SOTA X-CLIP Ni et al. (2022) on HMDB-51 by 2.5% in top-1 accuracy and loses 0.9% top-1 accuracy on UCF-101. Note that X-CLIP utilizes a more powerful video-specific prompting while we only use handcrafted prompting. The X-CLIP with handcrafted prompting gets 63.9% top-1 accuracy on UCF-101 which is 7.2% lower than ours.

## 5  CONCLUSION

In this work, we have presented triple 2D decomposition to efficiently implement a 3D ViT for spatiotemporal feature learning. The idea is simple yet effective. We believe that isolating the self-attention computation within each 2D data plane guides the network to focus on learning correlations among the most relevant tokens in a video clip. The proposed T2D block is a plug-n-play component, and it is implemented based on two SOTA image models known as CLIP and DaViT. Extensive evaluations are carried out on both two versions of T2D network. Very strong results are achieved across various benchmark datasets under both fine-tuning and zero-shot evaluation protocols. In the future, we plan to explore the self-supervised training of T2D networks on unlabelled video data, with an objective to fully exploit the temporal modeling capability.

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

# A  ARCHITECTURE DETAILS

**Patch embedding.** The patch embedding layer maps a video $X \in \mathbb{R}^{T \times H \times W \times 3}$ to a sequence of tokens $X_p \in \mathbb{R}^{N \times C}$, where H and W are the height and width of video frames, T is the clip length, 3 is the number of RGB channels, N is the number of embedded tokens, and C is the number of feature channels. It first divides the video clip into non-overlapping patches with the patch size of $P_t \times P_s \times P_s$ and use a linear layer to transform each patch to a feature vector. $P_s$ is set to 16 and 14 for base and large model which is the same as in CLIPRadford et al. (2021). When video is considered, we use $P_t = 2$ on SSv2 dataset to cover more frames and $P_t = 1$ on other datasets.

We do not use class token to aggregate spatial or temporal information because it does not bring accuracy improvement in action recognition task but increases inference time.

**Positional embedding.** We use space-time separable learnable positional embedding as shown in 1. A spatial positional embedding $e^s \in \mathbb{R}^{N_h \cdot N_w \times C}$ and a temporal positional embedding $e^t \in \mathbb{R}^{N_t \times C}$ are added on $X_p$. The spatial positional embedding is initialized from pre-trained weights and the temporal positional embedding is randomly initialized. In Eq.1, the spatial positional embedding, the temporal positional embedding, and the video patches are fused by a broadcast addition operation with proper reshaping. Concretely, we first reshape $X_p \in \mathbb{R}^{N \times C}$ from the shape of $N \times C$ to the shape of $N_t \times N_s \times N_s \times C$. The spatial positional embeddings $e^s$ is with the shape of $1 \times N_s \times N_s \times C$ and the temporal positional embeddings $e^t$ is with the shape of $N_t \times 1 \times 1 \times C$. We use the broadcast add operation to sum up $X_p$, $e^s$, and $e^t$, and then reshape the output back to the shape of $N \times C$.

**Spatiotemporal pooling.** We use a spatiotemporal pooling layer to aggregate patch-level outputs from T2D Transformer layer to video-level features. The spatial pooling is simply a global average pooling. In temporal pooling, we use an attentive pooling which is inspired from X-ViT Bulat et al. (2021). Given $N_t$ spatially pooled feature vectors $[z_0, ..., z_{N_t}]$ from $N_t$ "frames", we use temporal Transformer layers to aggregate temporal information. The self-attention in temporal Transformer layers are calculated across different frames. As the input is already pooled in spatial, it has negligible computational cost compared to the T2D Transformer layers.

**Classification head.** We use two types of classifier in our implementation. On SSv2, FineGym, and Diving-48, we use a learnable linear classifier which is randomly initialized and trained on the training set. On K400, UCF-101, and HMDB-51, we use an offline text-generated classifier to utilize the textual knowledge from CLIP's text encoder Wu et al. (2022). Take K400 as an example, we generate text embeddings of class names in K400 with the template of "a video of a person [CLASS NAME]". The generated text embeddings $\{w_1, ..., w_c\}$ are stacked to form a linear projection weight

$W \in \mathbb{R}^{c \times C}$, where $c$ is the number of classes and $C$ is the number of channels. Given the video-level feature vector $X_v \in \mathbb{R}^C$ from spatiotemporal pooling, the output of the classifier is calculate by:

$$Y = \sigma(W \cdot X_v), \tag{4}$$

where $\cdot$ is the matrix multiplication, and $\sigma$ denotes the softmax operation. The classifier is not optimized during training.

We find such offline text-generated classifier results in higher top-1 accuracy on K400 than randomly initialized linear classifier. The performance gain is about 0.2% ($84.5\% \rightarrow 84.7\%$) on the base model with 16 frames. We do not use this classifier in the ablation studies.

## B  DAVIT ARCHITECTURE

Dual attention Vision Transformer (DaViT) Ding et al. (2022) is a newly proposed hierarchical vision Transformer architecture for image. It utilizes two types of self-attention, one of which is calculated among local spatial tokens, and the other is calculated among channel tokens. The channel tokens are defined by inverting the channel dimension and spatial dimension of spatial tokens so that the attention on channel tokens is able to capture global context. We extend DaViT image models to DaViT-T2D video models by modifying its local spatial attention only. Similar to what we have done on ViT, we add additional XT and TY windowed attention after the original XY self-attention with skip connection. The window size for T is set to the clip length. We leave the channel attention unchanged as it captures the same global interaction as in image processing.

## C  RESNET ARCHITECTURE

To extend ResNet He et al. (2016) to a video model using T2D decomposition, we add XT ($3 \times 1 \times 3$) and TY ($3 \times 3 \times 1$) convolution after the XY ($1 \times 3 \times 3$) convolution in each ResNet Bottleneck block. Depth-wise convolution is used for the adding XT and TY convolutions. We do not modify the $1 \times 1$ convolution and the $3 \times 3$ convolution in the stem. We use ResNet-50 in our experiments and use the pre-trained weights from CLIP.

## D  DATASET DETAILS

**Kinetics-400.** Kinetics-400 (K400) Kay et al. (2017) is a large-scale video action recognition dataset collect collected from YouTube. Our Kinetics-400 dataset contains 234,584 training videos and 19,760 validation videos. We use 224 spatial input size in all experiments and sparsely sample 16 or 32 frames from the entire video. We use the same data augmentation and regularization as in X-CLIP Ni et al. (2022), including random horizontal flip, multi-scale cropping, color jitter, and random grayscale. We also use label smoothing, Mixup Zhang et al. (2018), and CutMix Yun et al. (2019) as regularization. We adopt the multi-view testing with four temporal clips and three spatial crops. We report the top-1 and top-5 classification accuracy on K400.

**Something-something-v2** Something-Something-v2 (SSv2) Goyal et al. (2017) is a large-scale action action recognition benchmark including 168.9K training videos and 24.7K validation videos over 174 classes. The labels are like "Pulling something from left to right", so capturing the motion is crucial to achieve high performance. We use segment based sampling Lin et al. (2019) to sample 32 frames from the original video. The augmentation and regularization strategy are adapted from MViT Fan et al. (2021), which includes repeated augmentation Hoffer et al. (2020), random augmentation Cubuk et al. (2020), random erasing Zhong et al. (2020), CutMix Yun et al. (2019), and Mixup Zhang et al. (2018).

**FineGym.** Finegym Shao et al. (2020) is a fine-grained action recognition dataset built on top of gymnastics videos. We use the same frame sampling strategy and data augmentation as it is in SSv2. The reported T2D-B model is with 32 frames and it uses the pre-trained weights from K400 because of the relative small scale of this dataset. Following Shao et al. (2020); Kim et al. (2021), we report the mean-class accuracy on two subsets of Gym288 and Gym99 that contain 288 and 99 action classes, respectively.

Table 9: Hyper-parameters used in K400.

| Model | Base | Large |
|---|---|---|
| Batch size | | 256 |
| Epochs | | 30 |
| Warmup epochs | | 5 |
| Learning rate | 1e-5 | 5e-6 |
| Learning rate schedule | | cosine |
| Optimizer | | AdamW |
| Weight decay | | 1e-3 |
| RandomFlip | | 0.5 |
| MultiScaleCrop | | (1, 0.875, 0.75, 0.66) |
| ColorJitter | | 0.8 |
| GrayScale | | 0.2 |
| Label smoothing | | 0.1 |
| Mixup | | 0.8 |
| CutMix | | 1.0 |

Table 10: Hyper-parameters used in SSv2, Diving48, and FineGym.

| Dataset | SSv2 | Diving48 | FineGym |
|---|---|---|---|
| Batch size | | 64 | |
| Epochs | 30 | 50 | 50 |
| Warmup epochs | | 5 | |
| Learning rate | 5e-5 | 3e-5 | 2e-5 |
| Learning rate schedule | | cosine | |
| Optimizer | | AdamW | |
| Weight decay | | 5e-2 | |
| Repeated augmentation | | 2 | |
| RandomAugment | | rand-m9-n4-mstd0.5-inc1 | |
| Random erasing | | 0.25 | |
| Label smoothing | | 0.1 | |
| Mixup | | 0.8 | |
| CutMix | | 1.0 | |

**Diving-48.** Diving-48 Li et al. (2018) is also a fine-grained action benchmark. It contains 18k videos with 48 diving action classes. As the background and the moving object is nearly the same across classes, the performance on this dataset heavily relies on effective temporal modeling. The setting for Diving-48 is the same as we use for FineGym. Following Bertasius et al. (2021), we report top-1 accuracy in Diving-48.

**UCF-101.** UCF-101 Soomro et al. (2012) is a video recognition dataset collected from YouTube. It includes 13,320 video clips with 101 categories. There are three splits of the test set and we report the average top-1 accuracy and standard deviation. We apply the zero-shot evaluation protocol and report results with 32 frames and a single view. The evaluated model is pre-trained on K400.

**HMDB-51.** HMDB-51 Kuehne et al. (2011) is a small dataset containing 7K videos with 51 categories. We use the same zero-shot evaluation protocol as we used for UCF-101.

# E  HYPERPARAMETER DETAILS

**Training Hyperparameters.** We present the hyper-parameters we used in different datasets and models in Tab.9 and Tab.10. Hyper-parameters on K400 are adapted from X-CLIP Ni et al. (2022) and hyper-parameters on SSv2 are adapted from MViT Fan et al. (2021). The learning rate shown in Tables are for CLIP initialized parameters. For randomly initialized parameters, we use a 100x learning rate for K400 and a 10x learning rate for SSv2, FineGym, and Diving48.

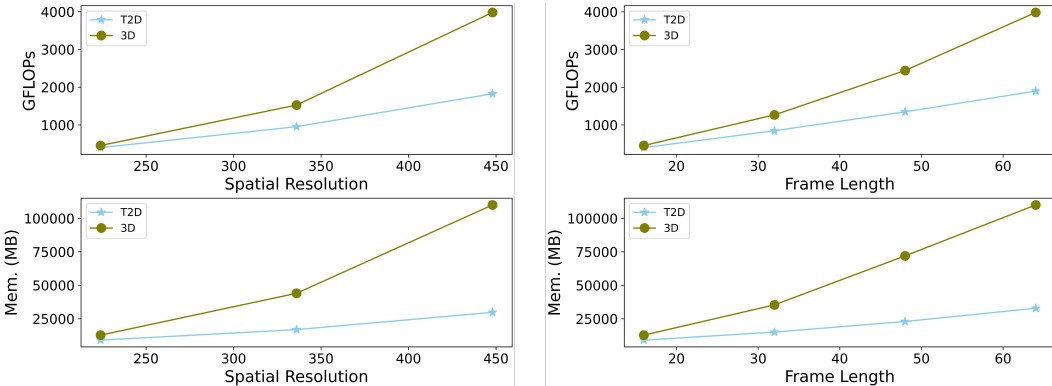

Figure 4: Empirical complexity analysis of T2D model versus 3D model. We plot the number of GFLOPs and the number of GPU memories as a function of spatial resolution (left), and frame length (right). As spatial resolution or frame length increases, our proposed T2D model leads to huge computational savings compared to the 3D model.

**Prompt templates in zero-shot experiments.** we use a single prompt template "a video of a person [CLASS NAME]" for HMDB-51, and multiple prompt templates for UCF-101. When multiple templates are used, their text embeddings are averaged so there is no additional cost compared to using a single template.

# F  ADDITIONAL EXPERIMENTS

**Empirical complexity analysis.** In addition to the theoretical complexity analysis in Tab. 1, we present additional empirical results to demonstrate the efficiency of the T2D model. As shown in Fig.4, the complexity advantage of T2D becomes obvious as spatial resolution or frame length increases. In particular, the 3D model will be out of memory with 64 frame length or 448 spatial resolution while our T2D model only needs about 30GB GPU memory under these settings.

# G  VISUALIZATION

In Fig.5 and Fig.6, we show the difference in attention maps among the 3D, 2D, 2D + 1D, and the proposed T2D attention. As we do not have class token in the network, we use the center pixel as the query. Compared to other methods, attention maps learned by T2D attention concentrate more on action regions.

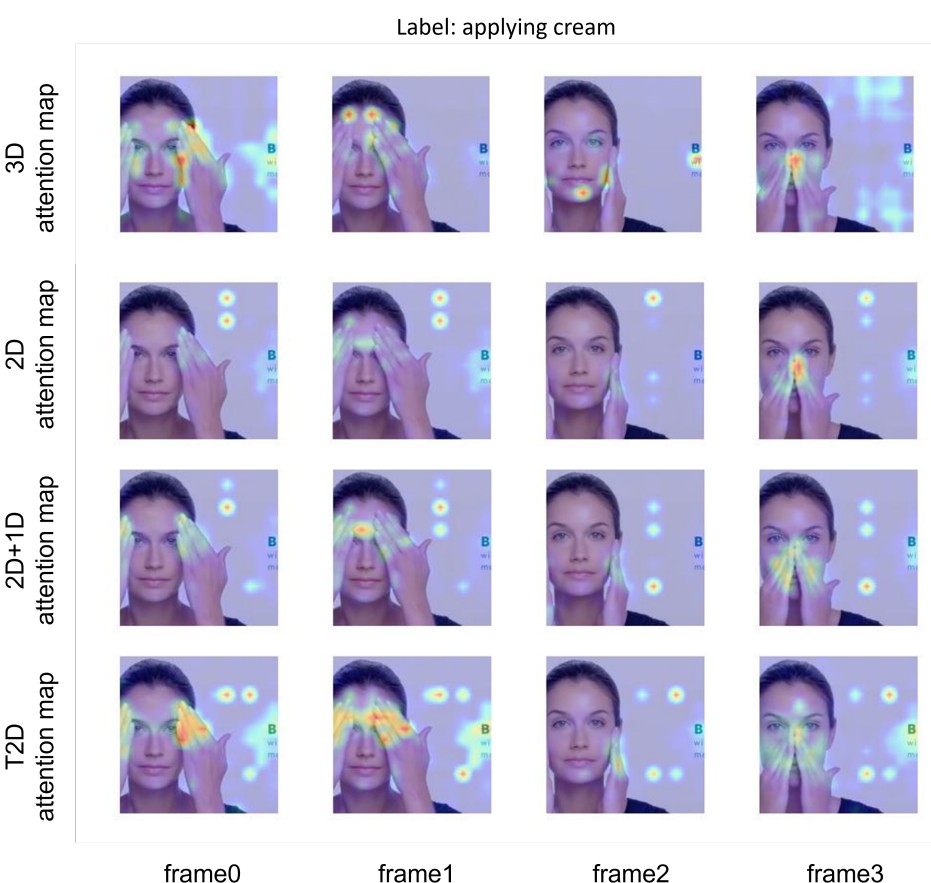

Figure 5: Visualization of attention maps on "applying cream". The 3D, 2D, 2D + 1D and T2D attention maps are provided in the top, medium, and bottom rows, respectively. T2D concentrates more on action regions.

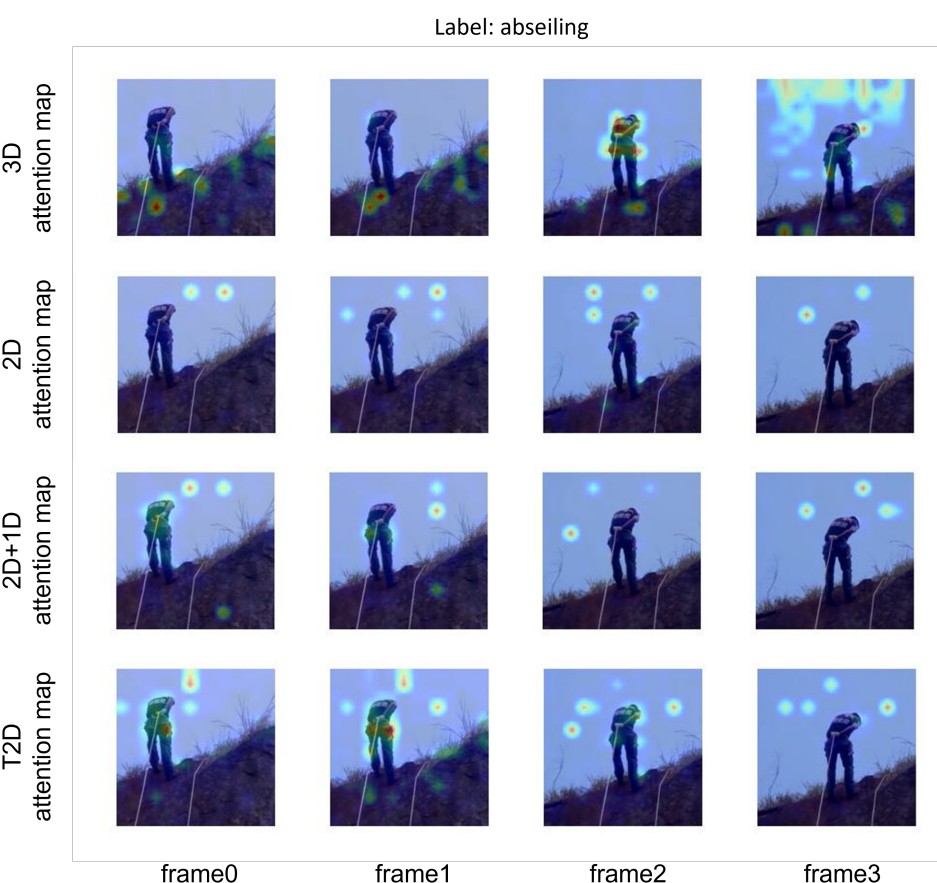

Figure 6: Visualization of attention maps on "abseiling". The 3D, 2D, 2D + 1D, and T2D attention maps are provided in the top, medium, and bottom rows, respectively. T2D concentrates more on action regions.

