# OpenReview forum: "T2D: Spatiotemporal Feature Learning Based on Triple 2D Decomposition"
_ICLR.cc/2023/Conference — Submitted to ICLR 2023_

### Official Review · Reviewer_LYJp · 2022-10-22

**Confidence:** 4
**Correctness:** 3
**Technical Novelty And Significance:** 2
**Empirical Novelty And Significance:** 2
**Recommendation:** 3

**Clarity, Quality, Novelty And Reproducibility:**

This proposed method original and writing is good. However, the novelty is very marginal.

**Strength And Weaknesses:**

Strength:
1. The proposed method has been validated on different datasets and the experiments are very comprehensive to validate the performance of T2D

Weakness
1. The idea of decomposing 3D attention into different planes of 2D attention is not novel, it has been proposed and experimented in R(2+1)D (cnn based architecture) and TimeSformer (transformer based architecture). The decomposition of T2D is very similar to axial decomposition in TimeSformer, which undermines the novelty of this proposed module

2. The performance on different datasets shows the introduction of T2D only leads to very marginal gain. For example, on Kinetics 400 and SSv2 datasets, T2D compared to regular 2D+1D, the performance gain is 0.4 and 0.3 respectively. Comparing proposed method with current SoTA method, when using pretrained CLIP-L ViT, the proposed method only achieved 0.0 and 0.3 in top1 and top3 accuracy, under similar GFLOPS cost.

**Summary Of The Paper:**

This submission proposed a new plug-in module which decomposes the 3D vision attention in video modeling. The proposed module, T2D, decomposes a self-attention among 3D spaces (spatial: XY and temporal: T) into three 2D space attention: XY, XT and YT. The proposed idea is very similar to what has been shown in R(2+1)D model and TimesFormer. The performance has been validated on different datasets including Kinetics 400, SomethingSomething v2, etc.

**Summary Of The Review:**

Based on the reviews in weakness part, I am prone to reject the submission.

---

> ### Author Response · Authors · 2022-11-18
> **Thanks for your valuable comments**
>
> Thank you for your valuable feedback. We have addressed your concerns and revised the paper by reflecting them as much as possible. Below are the detailed responses.
>
> __Q1: The idea of decomposing 3D attention into different planes of 2D attention is not novel, it has been proposed and experimented in R(2+1)D (cnn based architecture) and TimeSformer (transformer based architecture). The decomposition of T2D is very similar to axial decomposition in TimeSformer, which undermines the novelty of this proposed module__
>
> Answer: We thank the reviewer for the helpful comments expressing that the innovations and improvements over TimeSformer are not clearly described. While decomposition is a common idea, there are unique and non-trivial differences among ways to decompose 3D attention, each of which can yield very different results. In the original TimeSformer paper, the authors evaluated different decomposition methods, including 2D+1D (Divided Space-Time) and Axial. Note that the Axial decomposition in TimeSformer means decomposing 3D into three 1D fibers, not 2D slices. The following table from the TimeSformer paper shows that T2D has not been explored and that Axial decomposition is much inferior to Divided Space-Time (73.5 vs. 78.0 on K400). There are important distinctions between T2D and Axial in the Transformer decomposition domain. We have modified our Fig. 3 to make these distinctions clearer.
>
> | Attention           | Params | K400 | SSv2 |
> |---------------------|--------|------|------|
> | Space               | 85.9M  | 76.9 | 36.6 |
> | Joint Space-Time    | 85.9M  | 77.4 | 58.5 |
> | Divided Space-Time  | 121.4M | 78.0 | 59.5 |
> | Sparse Local Global | 121.4M | 75.9 | 56.3 |
> | Axial               | 156.8M | 73.5 | 56.2 |
>
>
> __Q2: The performance on different datasets shows the introduction of T2D only leads to very marginal gain. For example, on Kinetics 400 and SSv2 datasets, T2D compared to regular 2D+1D, the performance gain is 0.4 and 0.3 respectively. Comparing proposed method with current SoTA method, when using pretrained CLIP-L ViT, the proposed method only achieved 0.0 and 0.3 in top1 and top3 accuracy, under similar GFLOPS cost.__
>
> Answer: While we only slightly improve SOTA when using pretrained CLIP-L ViT, T2D offers significant gains on base and other smaller sized models which are more likely to be used in practice. For example, the Text4Vis method underperforms our method by 1.1% in top-1 accuracy with the base model size. In the additional experiment we include in Section 4.2 of the revised manuscript, we show that T2D consistently achieves non-negligible improvement of 0.4% top-1 accuracy gain over 2D+1D using two other network structures: DaViT and ResNet.
>
> In summary, although the performance improvement in the largest setting is not that significant, our method manages to achieve significant improvement in multiple other settings that are more likely to be implemented in practice. Therefore, as a general temporal module design, we believe the generalization capability of T2D is still plausible and novel in comparison to existing alternative method
>
> [TimeSformer] Gedas Bertasius, Heng Wang, and Lorenzo Torresani. Is space-time attention all you need for video understanding? In ICML, 2021

---

### Official Review · Reviewer_LefE · 2022-10-24

**Confidence:** 2
**Clarity, Quality, Novelty And Reproducibility:** Please see summary
**Correctness:** 3
**Technical Novelty And Significance:** 2
**Empirical Novelty And Significance:** 2
**Recommendation:** 5

**Strength And Weaknesses:**

Please see summary

**Summary Of The Paper:**

This paper proposes triple 2D decomposition (T2D) of a 3D vision transformer for spatiotemporal feature learning. It is achieved by decomposing the 3D representation into three 2D representations, e.g. XY, YT, XT. The isolated self-attention operation on three 2D representations improves the model performance on widely-used datasets.

Despite the performance improvements, this paper has the following weakness and shortcomings, which lead me to give a weak rejection.
1.	The decomposition operation is a bit too simple, and the improvement is not obvious from the results of the ablation study in Table2 compared with 2D+1D.
2.	I doubt the core difference between XT+YT in T2D and +1D in 2D+1D. The former will introduce more computation but bring little benefit.
3.	The decomposition operation does not bring efficient inference speed, nor does it greatly reduce the complexity.
4.	The ablation study on the combination of XY+XT, XY+YT, XY+XT+YT, etc. is missing. I think XY+XT or XY+YT contains enough temporal information for feature extraction.
5.	I think this decomposition operation applies to any vision network, either CNN or transformer. As the results in Table5, the performance improvement comes more from the network structure and pre-training, rather than the proposed method.


**Summary Of The Review:**

Please see summary

---

> ### Author Response · Authors · 2022-11-18
> **Thanks for your valuable comments**
>
> Thank you for your valuable feedback. We have addressed all the concerns and added missing comparisons in the revision. Below are the detailed responses.
>
> __Q1:The decomposition operation is a bit too simple, and the improvement is not obvious from the results of the ablation study in Table2 compared with 2D+1D.__
>
> Answer: Thanks for the comment and we understand your concern. We agree with you that the proposed T2D decomposition is a simple method, but it is effective across various settings, as it achieves consistent gain over 2D+1D decomposition. In the additional experiment, we experimented with two additional network structures, known as DaViT and ResNet. T2D consistently achieves non-negligible improvement of 0.4% top-1 accuracy gain over 2D+1D. Although the performance improvement at a single setting is not that significant, our method manages to achieve reasonable improvement at multiple settings. We believe that the generalization capability is an important advantage of our method.
>
> __Q2: I doubt the core difference between XT+YT in T2D and +1D in 2D+1D. The former will introduce more computation but bring little benefit.__
>
> Answer: This is a great point! We carried out two experiments trying to address your concern. First, in the additional experiment with the ResNet structure (details can be found in Section 4.2 and Table 2 in the revised manuscript), we implemented the 1D in 2D+1D and XT+TY in T2D with the light-weight depth-wise convolution. The resulting 2D+1D model and T2D model have almost the same GFLOPs. However, T2D model still manages to obtain 0.4% top-1 accuracy gain on K400.
>
> | Model (ResNet-based)   | GFLOPs | K400@1 |
> |----------|--------|--------|
> | T2D      | 93.6| 72.7   |
> | 2D+1D    | 93.2    | 72.3   |
>
> Second, we evaluated a 2D+1D+1D model, which uses one 2D spatial attention and two 1D temporal attention, based on the ViT network. As shown in the table below, the additional computation allocated to the 1D only brings 0.1% gain. Compared with the 2D+1D+1D model, T2D has less GFLOPs but gets better performance. Therefore, we conclude that the performance gain of T2D model is not from the increased computation but from the triple 2D decomposition mechanism.
>
> | Model    | GFLOPs | K400@1 |
> |----------|--------|--------|
> | T2D      | 397    | 84.5   |
> | 2D+1D    | 372    | 84.1   |
> | 2D+1D+1D | 462    | 84.2   |
>
> __Q3. The decomposition operation does not bring efficient inference speed, nor does it greatly reduce the complexity.__
>
> Answer: This is a great point! Thanks for giving us the opportunity to provide further explanations. The reduction in resources of the proposed decomposition operation will become obvious with increased video length and spatial resolution. We list the comparison in the following tables and plot them in Fig.4 (Appendix F) in our revision. Our T2D model leads to significant computational savings compared to the 3D attention model in terms of both memory cost and computational cost.
>
> | Frame Length | Resolution | Mem. (GB) |           | GFLOPs |             |
> |--------------|------------|-----------|-----------|--------|-------------|
> |              |            | T2D           | 3D        | T2D     | 3D         |
> | 16           | 224        | 9        | 13 (1.4x)| 397    |452 (1.1x) |
> | 16           | 336        | 17        |44 (2.6x) | 953    | 1526 (1.6x) |
> | 16           | 448        | 30       | OOM        | 1825    | 3982 (2.2x)|
>
>
> | Frame Length | Resolution | Mem. (GB) |           | GFLOPs |             |
> |--------------|------------|-----------|-----------|--------|-------------|
> |              |            | T2D       | 3D        | T2D    | 3D          |
> | 16           | 224        | 9         | 13 (1.4x) | 397    | 452 (1.1x)  |
> | 32           | 224        | 15        | 35 (2.3x) | 845    | 1266 (1.5x) |
> | 48           | 224        | 23        | 72 (3.1x) | 1346   | 2443 (1.8x) |
> | 64           | 224        | 33        | OOM       | 1898   | 3984 (2.1x) |
>
> OOM is short for out-of-memory.
>
> __Q4: The ablation study on the combination of XY+XT, XY+YT, XY+XT+YT, etc. is missing. I think XY+XT or XY+YT contains enough temporal information for feature extraction.__
>
> Answer: Great suggestion indeed! Following your suggestion, we have conducted the ablation experiments on the performance of XY+XT and XY+TY. We can see from the following table that removing TY or XT attention in the T2D model will result in decreased top-1 accuracy of 84.1% and 84.3%, respectively, showing that all three data planes contribute to the action recognition performance. We have included the results of this experiment in Section 4.2 in the revised manuscript.
>
> | Model          | K400@1 |
> |----------------|--------|
> | T2D (XY+XT+TY) | 84.5   |
> | XY+XT          | 84.1   |
> | XY+TY          | 84.3   |

---

> > ### Author Response · Authors · 2022-11-18
> > **Thanks for your valuable comments (continued)**
> >
> > __Q5.: I think this decomposition operation applies to any vision network, either CNN or transformer. As the results in Table5, the performance improvement comes more from the network structure and pre-training, rather than the proposed method.__
> >
> > Answer: Great point! To further demonstrate the generalization ability and performance improvement of T2D, we use the transformer backbone DaViT and CNN backbone ResNet and compare the T2D performance with 2D+1D baseline. As shown in the below Table, with the same pretraining strategy and backbone structure, our T2D consistently outperforms the 2D+1D baseline with non-negligible performance gain.
> >
> > |     Method    |     ViT       |               |               |     DaViT     |               |     ResNet    |               |
> > |---------------|---------------|---------------|---------------|---------------|---------------|---------------|---------------|
> > |               |     GFLOPs    |     K400@1    |     SSv2@1    |     GFLOPs    |     K400@1    |     GFLOPs    |     K400@1    |
> > |     2D        |     282       |     84.0      |     68.3      |     74        |     80.6      |     93        |     71.3      |
> > |     3D        |     452       |     84.3      |     67.2      |     92        |     80.3      |     186       |     70.9      |
> > |     2D+1D     |     372       |     84.1      |     70.2      |     86        |     80.6      |     93        |     72.3      |
> > |     T2D       |     397       |     84.5      |     70.5      |     91        |     81.0      |     94        |     72.7      |

---

### Official Review · Reviewer_NtS5 · 2022-10-25

**Confidence:** 3
**Correctness:** 3
**Technical Novelty And Significance:** 3
**Empirical Novelty And Significance:** 3
**Recommendation:** 8

**Clarity, Quality, Novelty And Reproducibility:**

Paper is well written and easy to follow. The method is novel and the experiments are comprehensive. Results are good -- improved upon SoTA.

**Strength And Weaknesses:**

Strength
- Decomposing ViT's 3D self-attention along three 2D plane is novel. It reduced computation complexity. This exact decomposition has not been explored before (see Fig 3), though similar to the "2D + 1D attention", and the "Axis attention".
- The authors show this simple model design can achieve competitive evaluation accuracy on video action recognition datasest, if not better.
    -- higher or competitive performance on Kinetics-400 and Something-Something-v2 benchmarks
    -- significantly improves SOTA accuracy on Diving-48, Gym99, and Gym288
- comprehensive ablation studies

Weakness
- It's not entirely clear by reading the abstract, the author decomposes the self-attention in 3D ViT. I suggest revise the abstract to make it clearer. Or consider move the Fig 2 to the first page.
- The improvement from 2D + 1D to T2D is not that significant (Table 2). Which is as expected - somewhat incremental.

**Summary Of The Paper:**

The paper proposed to decompose video data (3D, space 2D + time 1D) into 3 data planes. Each of the data plane is only 2D.

This allows the T2D model to create tokens in 3 2D planes separately. ViT 3D self-attention is decomposed along each plane. Compared with a single 3D self-attention, using the three 2D self-attention is less computationally expensive. The weights of the ViT's self-attentions are tied for the two 2D planes w/ time dimension.





**Summary Of The Review:**

The authors proposed an atomic and novel decomposition of ViT 3D self-attention along each 2D planes (XY, XT, and YT). The authors conducted extensive experiments to show this approach has advantage over existing approaches. The authors tuned the method well, so that it improved upon SoTA accuracy on a few popular action recognition datasets.

---

> ### Author Response · Authors · 2022-11-18
> **Thanks for your valuable comments**
>
> Thank you very much for recognizing the importance of our work. We are encouraged that the reviewer thinks our method is novel and our experiments are comprehensive. We also thank the reviewer for the valuable comments about the abstract part and have polished our paper based on the suggestions. Below are the detailed responses.
>
> __Q1: It's not entirely clear by reading the abstract, the author decomposes the self-attention in 3D ViT. I suggest revise the abstract to make it clearer. Or consider move the Fig 2 to the first page.__
>
> Answer: Thanks for the great suggestion! We have followed your suggestion and revised the second sentence in the abstract to “The idea is to decompose the self-attention operation in a 3D data cube into three self-attention operations in three 2D data planes.” We hope this revision makes the idea clearer.
> We are a bit unsure about whether you are suggesting us to move Fig.2 or Fig.3 to the first page. Please help advise and we are glad to make the adjustments in the final version.
>
> __Q2: The improvement from 2D + 1D to T2D is not that significant (Table 2). Which is as expected - somewhat incremental.__
>
> Answer: Thanks for this comment. In the additional experiment we include in Section 4.2 of the revised manuscript, we show that T2D consistently achieves non-negligible improvement of 0.4% top-1 accuracy gain over 2D+1D on two other network structures, known as DaViT and ResNet. Although the performance improvement at a single setting is not that significant, our method manages to achieve reasonable improvement at multiple settings. We believe that the generalization capability is an important advantage of our method.

---

### Official Review · Reviewer_Z88y · 2022-11-03

**Confidence:** 3
**Correctness:** 4
**Technical Novelty And Significance:** 3
**Empirical Novelty And Significance:** 2
**Recommendation:** 6

**Clarity, Quality, Novelty And Reproducibility:**

The paper writing should be improved
Novelty: Yes as commented above
Reproducibility: not sure

**Strength And Weaknesses:**

What's good:
1) First, the motivation of this paper is clear and intuitively effective. The idea of learning attention features by tri-plane modeling is novel and interesting.
2) The idea of triple 2D decomposition seems general and can be extended to related tasks.
3) Fig. 3 is really helpful for understanding the decomposition.
4) The proposed method shows better quantitative scores than the previous approach.

To be improved:
1) For Eq. 1, what does "+" means? Sum or concatenate?
2）Lacking the reference, discussion on previous triplane representation, and tensor decomposition methods. For example EG3D, TensoRF.

**Summary Of The Paper:**

The paper considers the action recognition problem from 3D video input. The core idea of the manuscript is to propose a triple 2D decomposition (T2D) for 3D vision transformer, which groups 3D video tokens along XY,XT and TY axis for self-attention computation. To address the memory cost issue in stander transformer models, the authors propose to extract spatial and spatiotemporal features from the 2D slices and fuse the attention feature for the final prediction. Extensive experiments on multiple datasets demonstrate that the proposed method is able to improve the recognition accuracy while greatly reducing the computation complexity.

**Summary Of The Review:**

The proposed triple 2D decomposition for action recognition from videos is novel and interesting to me, but the paper lacks a detailed discussion on the branch of decomposition work.

---

> ### Author Response · Authors · 2022-11-18
> **Thanks for your valuable comments**
>
> Thank you for acknowledging the novelty of our work and sharing the valuable feedback! Below, we answer the questions one by one.
>
> __Q1: For Eq. 1, what does "+" mean? Sum or concatenate?__
>
> Answer: Thanks for pointing it out. In Eq.1, “+” means broadcast add with proper reshaping. Concretely, we first reshape $X_p\in\mathbb{R}^{N\times C}$ from the shape of $N\times C$ to the shape of $N_t\times N_s\times N_s\times C$. The spatial positional embeddings $e^s$ is with the shape of $1\times N_s\times N_s\times C$ and the temporal positional embeddings $e^t$ is with the shape of $N_t\times 1\times 1\times C$. We use the broadcast add operation to sum up $X_p$, $e^s$, and $e^t$ and then reshape the output back to the shape of $N×C$. We have added a brief explanation in Section 3.1 and these details to Appendix.A in the revised manuscript.
>
> __Q2: Lacking the reference, discussion on previous triplane representation, and tensor decomposition methods. For example EG3D, TensoRF.__
>
> Great suggestion! We have added the following discussion in the introduction section of the revision: “Decomposition of higher-order tensors has found many applications in computer vision to deal with 3D geometry [EG3D, TensoRF] or video modeling [R(2+1)D, TimeSFormer]. Our work shares a similar idea as the tri-plane representation in EG3D to decompose a 3D tensor into three 2D data planes.”
>
> [EG3D] Eric R. Chan, Connor Z. Lin, Matthew A. Chan, Koki Nagano, Boxiao Pan, Shalini De Mello, Orazio Gallo, Leonidas J. Guibas, Jonathan Tremblay, Sameh Khamis, Tero Karras, and Gordon Wetzstein. Efficient geometry-aware 3d generative adversarial networks. In CVPR, 2022.
>
> [TensoRF] Anpei Chen, Zexiang Xu, Andreas Geiger, Jingyi Yu, and Hao Su. Tensorf: Tensorial radiance fields. In ECCV (32), 2022.
>
> [TimeSformer] Gedas Bertasius, Heng Wang, and Lorenzo Torresani. Is space-time attention all you need for video understanding? In ICML, 2021

---

> > ### Comment · Reviewer_Z88y · 2022-12-10
> > **Final Recommendation**
> >
> > Thanks for the reply, after reading the comments and responses from other reviewers and authors, I would like to recommend this paper with “marginally above the acceptance threshold”.

---

### Author Response · Authors · 2022-11-18
**General Response**

We thank all the reviewers for their insightful comments and suggestions. We are glad that the reviewers found our idea “novel and interesting” (Z88y, NtS5), and the method “shows better quantitative scores” (Z88y), “achieve competitive evaluation accuracy on video action recognition datasest, if not better” (NtS5), “improves the model performance on widely-used datasets.” (LefE), and “has been validated on different datasets” (LYJp). We appreciate reviewer NtS5 for pointing out that “This exact decomposition has not been explored before (see Fig 3), though similar to the "2D + 1D attention", and the "Axial attention".”

Nevertheless, the reviewers expressed their concerns that the improvement from 2D+1D to T2D is not significant (NtS5, LefE, LYJp) and whether the performance improvement comes from the network structure and pre-training or from the proposed method (LefE). In order to address this common concern, we carried out some additional experiments to evaluate different decomposition methods on two other network structures, namely CNN-based ResNet (CLIP pretrained) and Transformer-based DaViT (ImageNet-21K pretrained). We obtain very similar comparative results with the ViT model. T2D happens to achieve the same 0.4% top-1 accuracy gain over 2D+1D on K400 dataset when using the three different network structures with different pre-training. We argue that, as a general temporal module design, the generalization capability of T2D is still plausible and unachievable by any existing alternative methods.

The authors are very grateful to the reviewers for raising some other constructive comments about notation clarity, related work, and complexity reduction. We have answered all the questions raised, added the new experiments and analysis, and updated the submission in the system. The main changes, which are marked in blue in the revised manuscript, are:

•	Abstract: revised the second sentence to make the idea of “decomposes the self-attention in 3D ViT” clearer. (NtS5)

•	Section 1: mentioned the relationship with tensor decomposition and tri-plane decomposition in 3D geometry. (Z88y)

•	Section 3.2: modified the naming of different decomposition methods in Fig.3 to make the distinction between them more clear. (LYJp)

•	Section 4.2: added experiments to evaluate different decomposition methods on DaViT and ResNet and added experiments to compare T2D with XY+XT and XY+TY. (NtS5, LefE, LYJp)

•	Appendix.A: added the explanation of the “+” operation in Eq.1. (Z88y)

•	Appendix C: added the additional information of the ResNet architecture for the additional experiments.

•	Appendix F: added the empirical complexity analysis of T2D and visualize it in Fig. 4. (LefE)

Please help check the revision. We are happy to update the revision again if there is still any concern or suggestion. Below we will answer these questions in response to each individual reviewer.

---

### Decision · Program_Chairs · 2023-01-20

**Decision:**

Reject

**Justification For Why Not Higher Score:**

Reviewers who were part of the virtual meeting are just not that excited by this paper.  The AC feel that while the T2D decomposition is new, and could potentially be interesting, the experimental results (based on Tab 2 which was added in the rebuttal) do not show that the proposed decomposition is significant faster than the previously proposed 2D+1D (divided space time) decomposition from TimeSformer, or significant better performing.

**Justification For Why Not Lower Score:**

N/A

**Metareview: Summary, Strengths And Weaknesses:**

Summary: The paper proposes T2D, a new decomposition of video frames into three planes (a tri-plane representation).  The decomposition allows for more efficient self-attention with transformer-based architectures. Experiments compare with existing decompositions (none, space-only, divided space-time, axial, local-global) studied in TimeSformer, and show the proposed decomposition (for models without pretrained CLIP) has slight accuracy improvements and is more efficient in memory and compute compared to prior work.

Strengths
- The paper proposes a tri-plane decomposition of video frames which as not be studied in that domain
- The proposed decomposition performs well compared with prior work and is more efficient

Weaknesses
- The proposed decomposition is not that different from what was proposed in TimeSformer
- More careful comparison with 2D+1D (divided space-time) decomposition from TimeSformer show that the gain from the T2D decomposition is relatively small, and with similar computational cost.


**Summary Of Ac-Reviewer Meeting:**

The AC had a virtual meeting with reviewers LefE (rating 5), LYJp (rating 3), and Z88y (rating 6).  The slightly positive reviewer (Z88y) liked the paper but did not feel they were experts in the area.   Both reviewers LefE and LYJp felt the work lacked novelty as 1) the various decompositions were studied in TimeSformer and 2) the proposed decomposition is very similar to the axial and the divided space-time decomposition of TimeSformer.  The reviewers agreed that there was no other work that studied the specific proposed T2D decomposition but also did not find much value in it.  They felt the performance was not much better than prior work.  The AC noted that perhaps the performance was not that much better but perhaps it was more efficient.  In the end, reviewers remained unexcited about the work, indicating that it may be a better fit for a journal or some other conference.

Note that reviewer NtS5 (who is the most positive with a rating of 8) did not respond to emails, or enter discussion with reviewers or the authors.   The AC is thus discounting the reviewer's opinion.